# Living with psychosis in West and Southeast Africa: SUCCEED Africa's four-country situation analysis

Olubukola Omobowale[1] , Rachel Greenley[2], Grace Ryan[2], Olusegun Ogunmola[3], Lloyd Dzapasi[4], Abraham Jimmy[5], Anthony Sefasi[6], Mayowa Olusanmi[3], Rebecca Esliker[5], Alhaji Koroma[5], Adeola Afolayan[3], Rita Tamambang[3,8], Epiphania Munetsi[4], Janet Mambulasa[6], Ritsuko Kakuma[2] , Dixon Chibanda[4] , Olayinka Omigbodun[3,8] and Julian Eaton[2,7]

[1]Rehabilitative and Social Medicine Unit, Department of Community Medicine, College of Medicine, University of Ibadan, Ibadan, Nigeria; [2]Centre for Global Mental Health, London School of Hygiene & Tropical Medicine, London, UK; [3]Centre for Child and Adolescent Mental Health, College of Medicine, University of Ibadan, Ibadan, Nigeria; [4]Research Support Centre, University of Zimbabwe, Harare, Zimbabwe; [5]University of Makeni, Makeni, Sierra Leone; [6]Department of Clinical Psychology, Kamuzu University of Health Sciences, Blantyre, Malawi; [7]CBM Global Disability Inclusion, Laudenbach, Germany and [8]Department of Child and Adolescent Psychiatry, University College Hospital, Ibadan, Nigeria

## Research Article

**Keywords:**
*situation analysis*; *global mental health*; *psychosis*; *psychosocial disabilities*; *community-based rehabilitation*

**Corresponding author:**
Olubukola Omobowale;
Email: ocomobowale@com.ui.edu.ng

Olubukola Omobowale and Rachel Greenley are Co-first authors

Olayinka Omigbodun and Julian Eaton are Co-last authors

## Abstract

As part of the formative work of the SUCCEED Africa consortium, we followed a participatory process to identify existing gaps and resources needed for the development and implementation of a rights-based intervention for people with lived experience of psychosis in Malawi, Nigeria, Sierra Leone and Zimbabwe.

In 2021, we conducted a desk review of published and grey literature on psychosis in the four SUCCEED countries. Using an adapted version of the PRIME situation analysis template, data were extracted across the five domains of the WHO Community-Based Rehabilitation (CBR) Matrix: health, education, livelihoods, social and empowerment. This was supplemented with insights from personal communications with key stakeholders and the lived and professional experiences of team members.

Findings indicate that people with lived experience of psychosis have limited access to services and opportunities across the five CBR domains. Participation in social, religious, empowerment and political activities is restricted due to stigma and a lack of advocacy.

People with lived experience of psychosis in SUCCEED countries are not generally able to access support in line with essential components of CBR. There is a need for their greater inclusion in policy and advocacy activities.

## Impact statement

The World Health Organisation (WHO) promotes a person-centred, rights-based approach to mental health. While many global mental health projects undertake a situation analysis as their starting point for intervention development, these rarely take a rights-based perspective or include people with lived experience in the process. This situation analysis is tailored to cover five key domains of the WHO's Community-Based Rehabilitation (CBR) Matrix, and to involve people with lived experience of psychosis ("peer researchers") both as in-country team members and as members of a cross-site advisory group overseeing every stage of the process—from the adaptation of the situation analysis template to data extraction, synthesis, interpretation and dissemination. While traditional sources of grey and published literature consulted in desk reviews often omit anecdotal information, we also allow for lived and professional experiences to inform our situation analysis. The process and tools employed may prove informative for others seeking to develop rights-based interventions in similar contexts. Additionally, our findings may be of use to others working in the four countries under investigation (Malawi, Nigeria, Sierra Leone and Zimbabwe), as we report on a variety of factors such as human resources, policies, social and cultural influences, and sector-specific challenges that may impact implementation beyond the traditional metrics observed by policymakers and mental health programme developers.

## Introduction

### Mental health and psychoses in sub-Saharan Africa

The World Health Organisation (WHO) uses the umbrella term "psychosis" to describe conditions like schizophrenia and bipolar disorder (WHO, 2008). So-called "psychoses" are characterised by hallucinations, delusions and disordered thinking and behaviour (though it is worth noting that only a little over half of the people with bipolar disorder ultimately experience psychotic symptoms (Aminoff et al., 2022; Gaebel, 2012)). Over the past three decades, demographic trends have contributed to a significant increase in the Global Burden of Disease attributable to psychosis in Central and Western Africa (Charlson et al., 2018). In African psychiatric hospitals (Afolayan et al., 2015; Mwesiga et al., 2020) and in programmes scaling up decentralised mental health care (Cohen et al., 2011; Omigbodun et al., 2023; Ryan et al., 2020) psychoses often make up the majority of cases presenting to services, despite their relatively low prevalence (He et al., 2020).

In sub-Saharan Africa, formal mental health care has historically been provided by largely custodial, colonial-era asylums established under European rule (Heaton, 2021). Human rights observers in several countries have reported instances of forced restraint, prolonged seclusion, sexual assault and other abuses in health facilities, family homes, social care institutions and traditional and spiritual healing centres. Where maltreatment has been reported, political disenfranchisement and lack of appropriate representation, for example, through organisations of people with disabilities (OPDs), make it especially difficult for those who experience abuse to have their complaints heard and to seek justice (Bhugra et al., 2016).

### SUCCEED Africa: a rights-based perspective

The United Nations Convention on the Rights of Persons with Disabilities defines disabilities as "long-term impairments that interact with barriers in society to hinder full participation of affected persons" (United Nations, 2006). The disability rights-informed approach to mental health and psychosocial disabilities promoted by WHO requires multi-sectoral and multi-stakeholder action to tackle the various barriers and inequities faced by people with psychosocial disabilities (Funk et al., 2012; WHO, 2015; Hunt et al., 2022). The SUCCEED Africa (Support, Comprehensive Care and Empowerment of People with Psychosocial Disabilities in sub-Saharan Africa) Health Research Programme Consortium applies a disability rights lens to the development and evaluation of a community-based intervention for people with lived experience of psychosis in four countries of West (Nigeria and Sierra Leone) and Southeast Africa. (Malawi and Zimbabwe). SUCCEED follows a Theory of Change-driven development process aligned with the UK Medical Research Council (MRC) Framework for Complex Interventions (De Silva et al., 2014). This situation analysis is part of the formative research phase focused on identifying the existing evidence base as we begin to develop a theory of how the intervention will work within the local context(s), drawing on both published and grey literature as well as insights from team members with lived and/or professional experience in the countries under investigation.

### Previous situation analyses on mental health in sub-Saharan Africa

Situation analysis is a method of formative research that aims to understand from the earliest stages of intervention development the factors that may affect implementation, providing a comprehensive, evidence-based assessment of the situation in a given geographic area (Martin et al., 2016; Murphy et al., 2019). Previous situation analyses have examined mental health in specific countries (Abdulmalik et al., 2016; Hailemariam et al., 2019; Harris et al., 2020; Kauye et al., 2021; Mangezi et al., 2010; Olugbile et al., 2008), geographic regions (Esan et al., 2014) and partner sites of international research consortia(Hanlon et al., 2014; Mugisha et al., 2017). Increasingly, situation analyses are being used as part of a theory of change-driven approach to the design and evaluation of complex interventions in global mental health, generating initial insights into the assumptions underlying a "pathway of change" and how it is expected to work in a given context (De Silva et al., 2014) as well as identifying gaps in terms of mental health equity(Murphy et al., 2019). However, of the 24 situation analyses identified in Murphy et al., 2019 review, only three explicitly considered equity in their study design (Abdulmalik et al., 2016; Hailemariam et al., 2016; Mugisha et al., 2017) and none specifically mention a rights-based approach.

While several situation analyses have already been carried out in SUCCEED countries and are published as research articles, grey literature reports or country profiles, most are over 10 years old (Kauye et al., 2021; Mangezi et al., 2010; Olugbile et al., 2008). Some of the more recent publications have focused on particular cities (Oluwatayo et al., 2014) or topics, such as brain drain (Oladeji and Gureje, 2016) and have not taken a rights-based or person-centred perspective. A recent situation analysis from Sierra Leone (Harris et al., 2020) mentioned some efforts to draft legislation protecting the rights of people with mental health conditions; however, there is no further detail provided. Even where human rights or equity are considered (Abdulmalik et al., 2016; Esan et al., 2014; Hailemariam et al., 2016 Mugisha et al., 2017), people with lived experience of mental health conditions have not been involved in the design or conduct of the analysis–as is often the case in global mental health research (Ryan et al., 2019; Semrau et al., 2016). Hence, we sought to update and expand upon previous situation analyses following a more participatory process and examine a broader range of issues relevant to people with lived experience of psychosis in SUCCEED countries.

## Methods

We followed the secondary data review and analysis methodology described by Murphy et al. (2019). We combined data extracted from published and grey literature with information from personal communications with mental health experts and stakeholders, as well as insights from teams' personal and professional experience, to populate the domains of a structured situation analysis tool adapted from the Programme for Improving Mental Health Care (PRIME) study (Hanlon, 2014), described further below.

An advisory group comprising 18 people with professional or lived experience of psychosis (12 clinicians/researchers and six peer researchers) from across the four SUCCEED countries and the UK-based coordinating centre were responsible for overseeing the adaptation of the study tool, collecting information, reviewing the information collected, and contributing to the synthesis and interpretation of results. In keeping with SUCCEED's principles of co-production, peer researchers with lived experience of psychosis participated in the situation analysis throughout the entire process in each of the four country teams.

## Setting

This situation analysis was undertaken to inform intervention development. We therefore focused on the main study sites under consideration for inclusion in SUCCEED's future evaluation research: Mulanje and Phalombe (Malawi); Ibadan Southwest (Nigeria); Bombali and Koinadugu (Sierra Leone) and Harare and Mashonaland East (Zimbabwe). Relevant information was collected at the district (or local government area), regional and national levels.

## Measures

We revised the PRIME template to incorporate elements of WHO's Community-Based Rehabilitation (CBR) Matrix (WHO, 2015), as well as other resources and tools, namely: the WHO Mental Health Atlas (WHO, 2021); WHO Assessment Instrument for Mental Health Systems (AIMS) (World Health Organisation and Ministry of Health, 2006); and the WHO Quality Rights Toolkit (WHO, 2012). Additional items deemed relevant to the local context by the advisory group were also included in the situation analysis tool.

The working tool comprised 151 items spread across six sections. (Appendix 1) Where relevant, data were disaggregated by level of government (national, state and local government); health-care delivery (primary, secondary and tertiary); and education (primary, secondary or tertiary, public or private); as well as by socio-demographic characteristics (gender, rural versus urban). Table 1 outlines the sections of the situation analysis tool with examples.

## Data sources and extraction

Data for this situation analysis were extracted mainly from documents in the public domain, including peer-reviewed journal articles; grey literature including governmental and non-governmental reports, WHO publications and health surveillance data available through the WHO MiNDbank (http://www.mindbank.info) and Institute for Health Metrics and Evaluation (https://www.healthdata.org/). We extracted data across the five key domains of the CBR Matrix: health, education, social, empowerment and livelihoods (WHO, 2015). During the adaptation of the situation analysis tool, study teams were provided with a guidance note and online training highlighting common sources of relevant data (e.g., demographic and health surveys, national Department of Statistics reports, online sources, etc.), how to cite sources, and what to consider when selecting sources (hierarchy of evidence, original versus secondary sources, peer-reviewed versus grey literature, etc). Where there was no relevant data available from reliable sources of published or grey literature, teams incorporated their own insights based on personal and professional experience and personal communications with key stakeholders that had been carried out as part of local stakeholder engagement activities. (e.g., education professionals and peer researchers noted that people living with psychosis are typically excluded from school activities and experience discrimination in schools, though there was little formal documentation of this otherwise).

## Data collection and synthesis

Data were summarised and entered into the corresponding section of the situation analysis template for each country between March and July 2021. During this period, weekly online meetings

**Table 1.** SUCCEED situation analysis tool (see Appendix 1 for more details)

| Section(s) | Domain | Examples of items in the section |
|---|---|---|
| Background (26 items) | Socio-demographic, economic, health and disability indicators | Population size; ethnic groups; proportion of homes with electricity supply; adult literacy rate; life expectancy; infant mortality rate; proportion of people with psychosocial disability; proportion of people with psychosis, etc. |
| Health (76 items) | Promotion, prevention, medical care, rehabilitation, assistive devices | Mental health services; laws/ policies; mental health workforce pathways to mental health care; state of mental health facilities, etc. |
| Education (9 items) | Early childhood, primary, secondary and higher, non-formal, lifelong learning | School attendance and drop-out rates; opportunities for full participation in schools for students living with psychosocial disabilities; etc. education services; vocational training opportunities |
| Livelihood (14 items) | Skills development, self-employment, wage employment, financial services, social protection | Social welfare; financial and employment opportunities. Access to social welfare programmes; Access to financial opportunities; opportunities for employment, etc. |
| Social (12 items) | Personal assistance, relationships, marriage and family, culture and arts, recreation, leisure and sports, justice. | Social participation; shared decision making, legal supports. Opportunities for decision-making; opportunities for participation in social and religious activities; access to legal support in cases of abuse and discrimination, etc. |
| Empowerment (14 items) | Advocacy and communication, community mobilisation, political participation, self-help groups. Organisations of Persons with Disabilities. | Perceptions of people living with psychosis about the enforcement of their rights; opportunities for political participation; availability of support groups etc. |

were held among the study teams to monitor progress, highlight challenges and share experiences about the data collection process, making further edits to the template as needed.

Data synthesis was carried out over three stages. Study teams prepared individual country reports providing narrative description alongside the completed tables (available upon request). This was then followed by a cross-site peer review exercise in which each SUCCEED country team assessed another team's outputs for completeness, discrepancies and comprehensibility. Finally, country representatives synthesised the information as described in Figure 1.

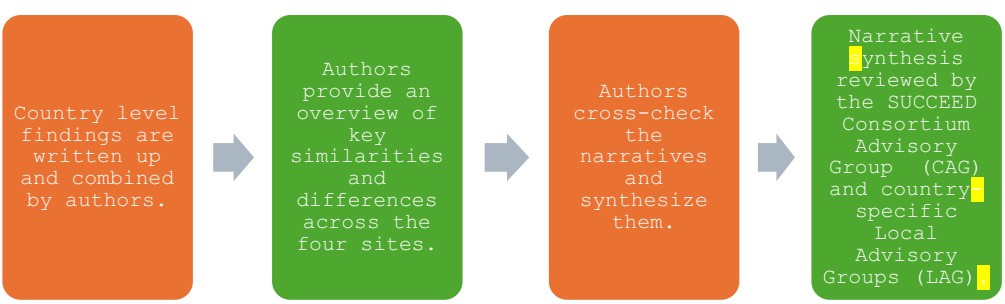

**Figure 1.** Synthesis flowchart.

## Results

### *Country profiles*

According to the World Bank classification system, Malawi and Sierra Leone are low-income countries, while Nigeria and Zimbabwe are lower-middle income countries. The classification relies on Gross National Income, which is only a proxy for overall development and may not adequately capture living standards or quality of life within a country. There is wide variation in country-specific demographic data, making generalisations about and across SUCCEED countries very difficult; however, Table 2 provides information on select social, demographic, economic, health and disability indicators at the national level, derived from country-specific Demographic and Health Surveys (DHS), Assessment Instrument for Mental

health System (AIMS) reports and Mental Health Atlas profiles from the WHO.

### *Health*

#### *Pathway to care*

Across all four sites, relevant literature indicates that people tend to consult faith-based providers for support when they experience psychiatric symptoms (e.g., in churches or mosques or from traditional healers) (Agara et al., 2008; Adeosun et al., 2013; Kajawu et al., 2019; Kokota et al., 2023). Anecdotal reports from team members suggested that formal health services are accessed either in addition to informal sources of support or after these sources have already been exhausted. In Malawi, specifically, peer researchers painted a more nuanced picture: people with psychiatric symptoms

**Table 2.** Selected sociodemographic indicators across SUCCEED countries

|  | Nigeria | Malawi | Sierra Leone | Zimbabwe |
|---|---|---|---|---|
| National Population | 200 m | 17.5 m | 7 m | 14.86 m |
| Population density | 217.5 | 186 | 97.2 | 37.32 |
| Ethnic diversity | Four major ethnic groups (Yoruba, Hausa, Igbo, Fulani) | Three major ethnic groups (majority in Mulanje and Phalombe are Lhomwe; Yao and Man'ganja are minority groups) | Two major ethnic groups (Temne and Mende) | Two major ethnic groups (Shona and Ndebele) |
| Language diversity | Three major languages (Yoruba, Hausa and Igbo) | Seven major languages (English, Chichewa, Tumbuka, Yao, Lomwe and Tonga, Sena) | Two major Languages (English and Krio) | Three major languages (Shona, Ndebele and English) |
| Religious diversity | Two major religions (Christianity and Islam) | Two major religions (Christianity and Islam) | Two major religions (Christianity and Islam) | Predominantly Christian |
| % Household with electricity | 66 | 86.2 | 23 | 85.7 |
| % Household with clean water supply | 59 | 9.5 | 34 | 77.1 |
| Life expectancy (years) female | 51 | 67.9 | 55.1 | 62 |
| Life expectancy (years) male | 47 | 61.5 | 53.47 | 63 |
| Proportion of people with physical disabilities | 13.1 | 10.4 | 34 | 7 |
| Prevalence of schizophrenia | 0.2% | Not available | Not available | Less than 1% |
| Prevalence of bipolar affective disorder | 0.1% | Not available | 1.4% | 0.5% |

in rural areas were expected to seek help first from traditional healers, while those in urban areas were considered more likely to present to health care professionals or faith-based healers.

### Duration of untreated psychosis

The duration of untreated psychosis (DUP, the period of time from onset of symptoms to initiation of treatment) ranges from 3 months in some Nigerian settings to 51 months in Malawi (Myaba et al., 2021). In Nigeria, the DUP was 25 months at the tertiary level (Adeosun et al., 2013). According to an anecdotal report from a Consultant Psychiatrist in a secondary health facility in Ibadan, Nigeria, DUP was much shorter: 3 months. In Malawi, the DUP was 51 months at both the secondary and tertiary levels of care (Myaba et al., 2021) This information was not available for Zimbabwe or Sierra Leone.

### Mental health promotion and prevention

Across the four country sites, some mental health promotion and prevention services and activities were in place, while others were non-existent. For example, peer researchers in Malawi, Nigeria and Zimbabwe reported that school-based mental health promotion and prevention activities like mental health clubs, awareness and anti-stigma programmes were in place at the district level, while parental and maternal mental health promotion and substance abuse prevention education were absent. Based on feedback from SUCCEED stakeholder engagement activities, teams also reported on the presence of secondary/tertiary prevention services and workplace mental health. Across the four study sites, some secondary/tertiary prevention services (early detection and prompt intervention, especially screening for mental disorders more generally but including psychosis and outpatient support) were present in tertiary hospitals only, with limited access. Workplace mental health promotion and prevention were lacking at all four country sites.

### Mental health system

In Sierra Leone and Malawi, mental health conditions including psychosis are managed at the secondary and tertiary levels of health care delivery, but inconsistently across districts and with non-governmental organisations often at the forefront. In Nigeria, psychoses are generally managed at the tertiary level, with some attempts to integrate mental health into primary care; however, staff competence, lack of political will and frequent transfer of trained staff are barriers to progress (World Health Organization, 2008). In Zimbabwe, primary care integration has been addressed through the Mental Health Gap Action Programme (mhGAP) training across seven towns and two cities (World Health Organization, 2008).

Perhaps reflecting this emphasis on specialist care, Nigeria had the largest public-sector mental health workforce: 0.12 psychiatrists, 2.41 psychiatric nurses, 0.12 psychiatric social workers, 0.07 clinical psychologists, 0.04 occupational therapists and 8.03 other health or mental health professionals per 100,000 population (World Health Organisation and Ministry of Health, 2006). Malawi, Sierra Leone and Zimbabwe had less than one mental health professional (all cadres combined) per 100,000 population (World Health Organization et al., 2009; Harris et al., 2020).

### Mental health financing

The total healthcare budget allocated for mental health ranged from less than 1% to 5% across the four sites. In Zimbabwe, 0.42% of the total healthcare budget is allocated to mental health, but most funds

**Table 3.** Mental health laws and policies in the SUCCEED countries

|  | Malawi | Nigeria | Sierra Leone | Zimbabwe |
|---|---|---|---|---|
| **Mental Health Policy (**Existence of an officially approved mental health policy) | Yes | Yes | Yes | Yes |
| **Mental Health Plan** (Existence of an officially approved mental health plan) | Yes | No | Yes | Yes |
| **Mental Health Law** (Existence of an officially approved mental health law) | Yes | No (at the time of data collection, the mental health bill had not yet been passed, it was passed in 2023 after data collection was complete) | Yes | Yes |

are used to maintain two psychiatric hospitals (Mangezi et al., 2010). Malawi had the second lowest allocation, with 1% of the healthcare budget allocated to mental health (Kauye et al., 2021). In Nigeria, this was 3% (World Health Organisation and Ministry of Health 2006), and in Sierra Leone there was no budget allocated to mental health (Harris et al., 2020).

### Mental health policy and legislation

Table 3 shows the existence of mental health laws and policies in the SUCCEED countries. The information provided was derived from Ministry of Health documentation from the respective study countries and supported by anecdotal reports from peer researchers.

## Education

### School drop-out rates

While deemed important for the situation analysis, school dropout rates for people with psychosocial disabilities were difficult to obtain. In Nigeria, the school drop-out rate for people with psychosocial disabilities in tertiary institutions was estimated at 20%, as reported by a social worker in a tertiary institution—though this is anecdotal information that could not be confirmed by the study team. School officials in Zimbabwe reported that 1.62% of the 34,808 total dropouts recorded in secondary institutions are related to mental health. This information was not available for Sierra Leone or Malawi.

### Discrimination in schools

Peer researchers noted that people with psychosocial disabilities experienced a range of barriers to participation in school activities, such as stigma, discrimination, social exclusion, poor school performance and in some cases decreased attention span and lower ability to understand social information like body language. In Nigeria and Sierra Leone, it was reported by school officials that students with psychosocial disabilities were not being given equal opportunities to occupy leadership positions and participate in social activities and decision-making. The school officials stated that they face discrimination, with students and teachers seeing

**Table 4.** Rights-based context of the mental health laws and policies by countries

|  | Malawi | Nigeria | Sierra Leone | Zimbabwe |
|---|---|---|---|---|
| Mental health law/policy |  |  |  |  |
| Promotes the participation of people with psychosocial disabilities in decision-making? | Yes | Yes | Yes | Yes |
| Recognises that people with psychosocial disabilities have the right to live on an equal basis with others? | Yes | Yes | Yes | Yes |
| Promotes transition towards community-based mental health services? | Yes | Yes | Yes | Yes |
| Allows for people with psychosocial disabilities to be subjected to medical treatment without their free consent? | Yes | Yes | Yes | No |
| Recognises that people with psychosocial disabilities enjoy legal capacity on an equal basis with others in all aspects of life? | Yes | Yes | Yes | Yes |
| Promotes a recovery approach? | Yes | Yes | Yes | Yes |

At the time of data collection, three SUCCEED countries had mental health acts that were obsolete (Sierra Leone, Nigeria and Malawi). Although Nigeria recently passed a mental health bill into law in 2023, the previous Lunacy Act of 1958 was still in place. In Zimbabwe, the existing Mental Health Act was enacted in 1996 and in Sierra Leone in 2010.

them as unfit to participate in intellectual activities like class debates.

### Access to education and vocational training

There do not appear to be any dedicated services tailored to support the inclusion of people with lived experience of psychosis in education and vocational training options across the four country sites, due to limited resources and a lack of understanding about the needs of people with psychosocial disabilities However, countries like Sierra Leone and Malawi are developing services to improve inclusive education. For example, the Malawi education sector analysis indicates an improvement in the training and recruitment of special needs education teachers to provide tailored learning programmes for people with intellectual disabilities, though coverage is limited, and it is not clear if this would include people with psychosocial disabilities, such as those with lived experience of psychosis (UNICEF and Ministry of Education, 2019).

### Livelihood

There was little information on livelihoods in either the grey or published literature, but peer researchers described a general urban/rural divide in terms of access to livelihood opportunities for people with psychosocial disabilities. Opportunities in rural settings appeared to be limited (e.g., to agriculture), without access to other skills acquisition (e.g., digital/computer literacy) which are often available in urban areas. However, peer researchers in Zimbabwe noted that access to skills development is also poor in urban settings due to stigma. Peer researchers in Sierra Leone also observed that

there are special projects for livelihood development and business start-ups for people with disabilities funded by the government through the Ministry of Social Affairs, but access for people with psychosocial disabilities is not specifically protected.

### Social participation

Peer researchers in Malawi, Zimbabwe and Sierra Leone reported that regardless of setting (rural/urban), people with psychosocial disabilities have very little choice about their needs for personal assistance (e.g., self-care and other daily functions) or about who might provide this assistance. In contrast, peer researchers from Nigeria felt that there was more autonomy, particularly in urban settings, though important life decisions (such as who to marry) were still often left to trusted family members, reflecting wider societal practices. In Zimbabwe and Sierra Leone, peer researchers reported that there was greater autonomy in decision-making surrounding relationships and family life, compared to Nigeria. The Marriage, Divorce and Family Relationship Act (2015) prohibits people with mental and intellectual disabilities from getting married in Malawi; however, according to our peer researchers, the extent to which this is actually enforced is unclear.

In both rural and urban settings across the four country sites, people with psychosis received social support in the form of food donations, financial assistance, counselling, prayer and referral to health services from religious institutions. Religious institutions generally allowed people to participate in religious activities, except in Sierra Leone where prejudice was considered to be especially prevalent in both rural and urban settings.

### Empowerment

In Sierra Leone and Zimbabwe, it was reported that the policies in the country do not provide people with disabilities equal rights in all settings. In Nigeria, though laws on the rights of people with disabilities exist (e.g., the Prohibition of Discrimination against Persons with Disabilities Act, 2019 and the National Mental Health Act, 2021), they are rarely respected or enforced.

In two of the country sites (Sierra Leone and Zimbabwe), people with mental health diagnoses are not allowed to vote or to be members of a political party. Furthermore, stakeholders in Sierra Leone explained during engagement activities that they are neither eligible to be members of political or civil organisations nor to run for political office.

Although recognised Organisations of Persons with Psychosocial Disabilities (OPDs) exist across all four sites, they are small and have low coverage and influence. It was noted by peer researchers in Nigeria that people with psychosocial disabilities are not aware of their existence, and OPDs do not generally have the resources to adequately represent the interests of people with psychosocial disabilities. Further, in cases of abuse, people with psychosocial disabilities are not able to seek justice.

## Discussion

### Key findings

This situation analysis documented similarities and differences across four sites in sub-Saharan Africa, across the various domains of the CBR Matrix: health, education, livelihood, empowerment and social. Key findings that may be especially relevant to the development of SUCCEED's community-based intervention

include: the high levels of discrimination and near-absence of any sort of accommodation for people with psychosocial disabilities in mainstream education; the limited opportunities for other forms of education and livelihoods, such as vocational training and skills development, particularly in rural areas; social exclusion, lack of political will and poor access to decentralised health care. The current level of health care provision in the lower two levels of the health system is weak, but policies exist that in principle should enable access to decentralised services. Exclusion is even more rampant in other sectors such as education, social welfare, housing and employment. While some countries have special projects for livelihood development and business start-ups for people with disabilities, access for people with psychosocial disabilities is not specifically protected, and stigma remains an important barrier. Ultimately, Organisations of Persons with Psychosocial Disabilities are much weaker than those for other disabilities across SUCCEED sites and thus are not able to adequately represent their interests or obtain justice in cases of human rights violations. Below, we discuss these key findings in the context of the broader literature on psychosis and psychosocial disabilities in sub-Saharan Africa and other low- and middle-income country settings.

## Health

Reducing DUP is one strategy for improving outcomes of psychosis, and having first contact with general hospitals is associated with shorter DUP (Sankoh et al., 2018.) In our study, the shorter DUP reported by secondary health facilities underlines the importance of having mental health care that is as close to the people as possible. Unfortunately, across all SUCCEED country sites, except in Nigeria where some efforts are being made to integrate mental health care into primary care, psychoses are not often managed at the primary health care level, despite efforts to scale up mental health programmes, for example in some states of Nigeria (Abdulmalik et al., 2016) This is a pervasive challenge in the African region, though governments of some countries such as Cape Verde, Kenya, Mozambique, Rwanda and Uganda are making progress (Ryan et al., 2020). It is equally concerning that there remains a huge shortage of mental health professionals in all four participating countries, with poor morale and migration making the situation worse across Africa (Oladeji et al., 2016). Although comparable to the African regional average of 1.4 per 100,000 population, the mental health workforce figures in SUCCEED countries are far lower than the global average of 9 per 100,000 population (WHO, 2018).

Rehabilitation services were present to some extent (at least at the secondary health care level) in all the participating countries except Sierra Leone. However, the rehabilitation services reportedly available were inadequate, as most of the skill-acquisition opportunities did not prepare people well for the job market. Younger people in particular favoured information and communication technology (ICT)-related skills. This presents an opportunity for SUCCEED as it looks to develop a CBR intervention that would be consistent with recovery approaches emphasising personal choice.

In all four participating countries, the proportion of the total health budget allocated to mental health was much lower than the 15% recommended by the WHO and the Abuja Declaration (2006). However, funding for mental health care should not be borne by the health sector alone, and there is a need for a framework for effective intersectoral collaborations and proper mapping of alternative sources of funding for mental health care (Iemmi, 2019).

## Education

Several studies in low- and high-income countries suggest that poor mental health contributes to high school drop-out rates (Andersen et al., 2021; Hjorth et al., 2016; Humensky et al., 2019; Lawrence et al., 2023; Weybright et al., 2017) because problems often begin in late adolescence or early adulthood, and many have difficulty completing high school and entering postsecondary education. In our study, information on school drop-out rates among people with psychosocial disabilities was largely unavailable across the four country sites. Anecdotal reports from Nigeria reported an estimated 20% school dropout rate and Zimbabwe reported 1.62% specifically for people with psychosocial disabilities, but the quality of this data was very poor. However, this dropout rate is much lower compared to findings from a study of an urban, disadvantaged, African American sample of people with psychosis (Goulding et al., 2010), which reported a 44% school dropout rate. This finding was thought to be a marker of numerous social problems, indicating a potential point of intervention to enhance long-term psychosocial functioning. Supported education is one such intervention that researchers have begun to successfully tailor and apply to individuals living with psychosis in high-income countries (Falkum et al., 2017; Killackey et al., 2008).

Our study identified no professional or other vocational training programmes for people with psychosocial disabilities across the four sites. Malawi has a special education programme, but it is unclear whether psychosocial disabilities fall under its remit (UNICEF and Ministry of Education, 2019).

## Livelihood and social protection

People living with psychosis often lack basic needs such as food, clothing and money, in the context of high unemployment rates and inadequate social protection (Ofori-Atta et al., 2010)). In all four countries, support in the form of food or financial assistance was given through charity models (often by religious institutions), rather than being seen as a right.

Without concerted efforts to support development of livelihoods, people with psychosocial disabilities are more likely to remain marginalised (Funk et al., 2012). The WHO includes psychosocial strategies that enhance vocational and economic inclusion in service planning and delivery recommendations (Kakuma et al., 2017). However, there are few examples of livelihood programmes that are integrated into mental health services, despite livelihood often being considered central to progress in other social domains. Basic needs (WHO, 2008) for example provides an established model of comprehensive mental health service provision with a livelihood component but does not operate in SUCCEED countries. There is evidence from high-income countries that individual placement and support appears to be the best approach (Rinaldi et al., 2008) however this has been contested (Pichler et al., 2021), and the applicability of this model of intervention to low- and middle-income settings is unclear.

## Empowerment

One of the most pressing issues in research on psychosocial disabilities is the under-involvement of people with lived experience and their families (Bauer et al., 2019). To be empowered, an individual must have choice, influence and control in their lives (Funk et al., 2006; Murphy et al., 2021). While user-led research and co-production have played important roles in influencing health care and human rights reform in some high-income countries (Laugharne and Priebe, 2006), this is less common in low- and middle-income countries (Ryan et al., 2019; Semrau et al., 2016).

In SUCCEED countries, people with psychosis and their families have not been involved as equal partners in mental health decision-making processes historically, perpetuating social exclusion and discrimination across many facets of life (Sugiura et al., 2020). In Sierra Leone and Zimbabwe, we found laws prohibiting people diagnosed with psychosis from running for office, dividing decision-makers from service users (Toguem et al., 2022). Anecdotal reports from peer researchers suggested that people with a lived experience of psychosis are routinely prohibited from making their own decisions, in both rural and urban areas. Unequal power dynamics between health practitioners and people using services may also contribute to coercive decision-making in these settings (Sugiura et al., 2020).

Similarly in research and policy-making, "experts by experience" are best positioned to define their needs and how to address them, with support from "experts by profession." The emergence of recognised OPDs can facilitate this, but while OPDs exist in SUCCEED countries, they are not well known. As OPD presence strengthens, the use of disability rights perspectives can promote the process of empowerment (OHCHR, 2008). At the community level, people affected and their families should be encouraged to form associations and actively promote their rights and well-being.

### Strengths and limitations

This situation analysis benefits from the involvement of people with lived experience and professional experience, respectively, in the countries under investigation. As we discovered in the process of completing our desk review, there was very little literature available on any of the CBR domains beyond health. Consequently, we often had to rely on our own team members and their personal communications with key stakeholders to help "fill in the blanks". However, our reliance on largely anecdotal information could also be considered a weakness. Further, the lack of standardised information made it difficult in some cases to meaningfully compare the situation across SUCCEED countries. Finally, the data for this study were collected in 2021 and so may not fully represent the current situation in all the SUCCEED country sites, though the general lack of timely, representative data on psychoses is a general issue in this region (Omigbodun et al., 2023). Many of the challenges relating to mental health care (e.g. poverty, shortage of manpower, inequities in access, scarcity of community-based care, etc.) persist and may have worsened following the coronavirus disease 2019 pandemic. This desk review will be followed by further qualitative studies as part of SUCCEED's formative research phase, which will offer a more nuanced exploration of the lived experience of psychosis in these four countries, from the perspectives of a range of different stakeholders. Country-level reports are also available upon request for future research and intervention development.

### Conclusion

The SUCCEED situation analysis revealed a lack of rights-based mental healthcare, as well as a general lack of educational, livelihood, social and empowerment opportunities for people with psychosocial disabilities generally and for those with lived experience of psychosis, specifically, across four African sites. High levels of stigma and poor community awareness of psychosocial disabilities remain major barriers to full and equal participation. There is a need for greater inclusion of people living with psychosocial

disabilities in policy and advocacy activities. Taking a comprehensive, evidence-based and contextually appropriate approach enables the identification of the main gaps, needs and resources. In the case of SUCCEED Africa, a commitment to recognising the barriers faced by people with psychosocial disabilities requires a multi-sectoral, multi-stakeholder approach to address these inequities. This research sheds light on the specific challenges faced by individuals with psychosocial disabilities, which in many cases were similar across otherwise diverse contexts. Ultimately, this will be a valuable tool in planning the next phase of intervention development and evaluation for the SUCCEED programme, and a similar approach could be used by others as part of a process of developing inclusive interventions to make health and other sectors more responsive to the needs of people with lived experience of psychosis.

**Open peer review.** To view the open peer review materials for this article, please visit http://doi.org/10.1017/gmh.2024.122.

**Supplementary material.** The supplementary material for this article can be found at http://doi.org/10.1017/gmh.2024.122.

**Data availability statement.** Data for this situation analysis can be made available upon reasonable request.

**Acknowledgements.** We wish to acknowledge the collaboration of officials of various Ministries of Health, Education and Social Welfare in all the SUCCEED countries. We also acknowledge additional members of the SUCCEED study teams who contributed to the adaptation of the situation analysis tool and to the information collected: (Thomas Shakespeare, Tolulope Bella-Awusah, Haleem Abdurahman, Olayinka Aturu, Bisola Fasoranti, Olayinka Bamidele, Musa Buyanga, Tichaona Gumunyu, Nyardzo Goba, Philani Kinyabo).

**Author contribution.** All authors contributed to the conceptualization of this paper and to the process of revising it for intellectual content. JE and O Omigbodun provided the technical oversight and supervision for this cross-site situation analysis paper. O Omobowale, RG, GR, LD, AS, O Ogunmola, AK, EM, MO, RE and JM coordinated the data extraction, analysis and synthesis. O Omobowale, RG and GR led the development of the first draft of the manuscript as well as revisions. JE, LD, RT, AS, IJ, RE, AA, AK, EM, RK and O Ogunmola also provided technical inputs for the revision and editing of the manuscript. All authors read and approved the final manuscript.

**Financial support.** This project was funded by UK aid through the Foreign, Commonwealth and Development Office (FCDO), United Kingdom (EPPHZT7410), however, the views expressed in this manuscript do not necessarily reflect the UK Government's official policies. The funders had no role in the preparation of the manuscript or the decision to publish.

**Competing interest.** The authors declare no conflict of interest.

**Ethical consideration.** This was a desk review of published and grey literature; ethical approval was not required.

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
