## [Reviewer Report]

Title: Developing a rights-based Intervention for people with a lived experience of psychosis: situation analysis in four African countries.

The paper would be a valuable and factual resource for professionals, practitioners, and policy makers; as well as people with lived experience of psychosis and their family caregivers, to stimulate the development of appropriate intervention programmes for people with a lived experience of psychosis.

The introduction addressed the background information on the subject, while the methodology is appropriate for the objectives of the work.

Results:

There are no in-text citations on sources of data presented in the results section - under health, education, livelihood, social participation, and empowerment. This is necessary because information presented did not originate from the authors, as in observational situation analysis, contributing to the credibility and integrity of the work.

The authors admitted that information on pathway to care was mainly derived from peer researchers’ and care providers’ insight. This would not apply to Nigeria and Malawi, in view of the available published literature on the subject.

A general statement under discussion that “psychoses are not managed at the primary health care level across all SUCCEED country sites” would be a non acknowledgement of isolated efforts of a statewide mental health integration programme, in Nigeria, where psychoses is managed at the primary care level.

The authors should also ensure adherence to acceptable formats for in-text and reference section citations for both published and grey literatures.

---

## [Reviewer Report]

Situation analysis is an established approach to understand context and a useful first step in formative research. Using the CBR framework as a structure is novel and highly relevant given the ultimate aim is to develop a community-based intervention for psychosis. However there are several weaknesses in the reporting, detailed below.

Methods

- Can you briefly explain how you identified relevant peer-reviewed journal articles and grey literature?

- You state that study teams were given guidance on what to consider when selected sources ‘e.g. peer-reviewed vs grey literature’. Can you give more information on what the guidance was e.g. were both peer-reviewed and grey literature reported if both were identified, or was peer-reviewed literature given precedence? Was anecdotal evidence used only if peer-reviewed/grey literature was not available? etc

- Did you take any steps to establish the ‘trustworthiness’/ accuracy of stakeholder information e.g. by triangulating against other sources

- It is not very clear at what level- district/national etc- was being sought- did this differ by domain? Were a priori decisions made as to which level would be sought for which domain, and if so what was the rationale?

- The scope of the included information is not always clear- in some sections it is focused on people with lived experience of psychosis, in others on psychosocial disabilities more broadly. It would be good to be clearer when specific information on psychosis is not available, and the relevance of the broader perspective.

- The results rely heavily on information from stakeholders. It would be good to see more detail on who the ‘clinicians/researchers’ are so the reader can better judge how complete their knowledge of the context is e.g. would the group necessarily know about all instances of maternal mental health promotion, school-based programmes etc. Where were the gaps e.g. district health officials, policy makers? Education vs health; Voluntary vs governmental sector

- Can you add the situation analysis tool as a supplementary file

Results

- Overall: To make it easier to compare information between sites and see where the gaps are I suggest creating table/s summarizing all data.

- Table 1- it would be helpful to reference the sources, and potentially state the year, within the table so the reader can get a sense of how comparable the data is

- There is some peer-reviewed literature on pathways to mental healthcare for people with psychosis in Nigeria and Malawi. It would be good to see this included as well or instead of the anecdotal information

- Duration of untreated psychosis- it would be helpful to see the source/s and year of this data to help understand its validity, relevance and comparability

- Mental health promotion and prevention. It is unclear whether these were considered at a district/ national level. It would also help to know the relevance of these services and activities (e.g. childhood stimulation programmes) to psychosis.

- ‘secondary/tertiary prevention services (early detection and prompt intervention, especially screening and outpatient support)’- could you clarify the types of mental health condition/s this relates to- is this screening for psychosis? If not, suggest clarifying the relevance to psychosis

- Table 3 Mental health laws and policies- ‘Existence of an officially approved draft mental health document’- It is unclear what ‘document’ refers to. Is this row needed? Suggest adding year of policy/plan/law and reference the documents where possible.

- Simply knowing the policy/ plan/ law exists is not very informative in isolation. It would be more interesting to know to what extent they reflect the CRPD and/or address the CBR pillars in relation to psychosis. Peer-reviewed literature analyzing this question could be drawn upon https://pubmed.ncbi.nlm.nih.gov/35772283/

- ‘In rural and urban settings of Malawi, Zimbabwe, and Sierra Leone, peer researchers reported limited choice about personal assistance needs and who provides this assistance. However, in urban settings in Nigeria, greater autonomy was reported’ . Could you explain what ‘personal assistance’ might mean in relation to psychosis, as well as what ‘greater autonomy’ means in this context.

- “In Sierra Leone and Zimbabwe, it was reported that the policies in the country do not provide people with disabilities equal rights in all settings. In Nigeria, though laws on the rights of people with disabilities exist, they are not respected or enforced”- Could you clarify which policies and laws are being referred to here? To what extent does this relate to people with psychosis (or people with psychosocial disabilities)? What settings are being referred to?

Discussion

- Can you comment on how you anticipate the findings will shape your theory of change and/or your intervention design?

- Suggest adding to the limitations that data was collected in 2021- given this is a situation analysis, can authors comment on extent to which findings are still relevant?

References

- Some references are repeated in the bibiography

Other minor comments

Introduction

-“people with lived experience of a mental health condition were not involved in the design or conduct of the analysis – as was often the case in global mental health research”- suggest amend to ‘as is often the case..’

Methods

- ‘to populate the domains of a structured situation analysis tool adapted from the Programme for Improving Mental Health Care (PRIME) study (PRIME 2013).’- change citation to Hanlon 2014

Results

- ‘mhGAP’ acronym needs expanding

- Livelihood – ‘without access to other skills acquisition (e.g., digital/computer literacy) as in urban areas.’ – is it the case that computer literacy training is available for people with psychosocial disabilities in urban areas? This could be made clearer

-

Discussion

- ‘In addition, caregivers find it very difficult to provide support due to a lack of finances, time, and social support(Siskind and Yung 2022)’- this reference does not seem to directly relate to the text.

---

## [Reviewer Report]

I thank the authors for this piece of work. However, there are some minor but important revisions that need to be done.

1) The manuscript generally requires some thorough editing.

2) Many of the references are old. It would be nice to have references not older than 10 years.

3) You mention involving 18 persons with professional and lived experience across the 4 countries. Can you give the breakdown/distribution per country.

4) There is some contradiction under “pathway to care”, page 7. You start by stating that the first port of call are family and friends; but later state that in Malawi, they first seek care from traditional healers. Please clarify.

5) You seem to use the terms “psychosocial disabilities” and “psychosis” interchangeably, as though they are synonymous. This should be put right.

6) The content under results and discussion sections doesn’t seem to match with the title of the manuscript. It is more of a situation analysis. There isn’t much in line with developing the intervention, as the title seems to suggest. The authors should either change the title or add content on developing the intervention, in order to improve the manuscript.

---

## [Reviewer Report]

Please could the authors kindly indicate in their response the location e.g. page and line number, of amendments to the manuscript. And ideally include a tracked changes version so it is clearer what amendments have been made. In the current format it is hard to locate the relevant edits.

---

## [Reviewer Report]

The authors have addressed the concerns earlier raised. However, the manuscript still has some grammatical and typing errors. The authors should work with the editorial team to polish it further.

---

## [Editor Report]

Please set the settings of the track changes fucntion to show “All markup”. Further, there are minor grammatical and editing issues throughout, kindly address and re-submit a proofread document.

---

## [Reviewer Report]

It is hard to review these revisions without a tracked changes or highlighted version of the revised manuscript. Several of the comments have been addressed in the text. However whilst the authors have given responses to Methods comments 1,2,3,4 and 6, there have not been any edits to the text, that I can see. (For comment 6 it would make sense to include a sentence in the text as well as the supplementary table). I also cannot see new text relating to Discussion comment 1.

I am happy to recommend accept if the editor is happy with these omissions.